# UltraRE: Enhancing RecEraser for Recommendation Unlearning via Error Decomposition

**Yuyuan Li**
College of Computer Science
Zhejiang University
Hangzhou, China
11821022@zju.edu.cn

**Chaochao Chen**[*]
College of Computer Science
Zhejiang University
Hangzhou, China
zjuccc@zju.edu.cn

**Yizhao Zhang**
College of Computer Science
Zhejiang University
Hangzhou, China
22221337@zju.edu.cn

**Weiming Liu**
College of Computer Science
Zhejiang University
Hangzhou, China
21831010@zju.edu.cn

**Lingjuan Lyu**
Sony AI
Japan
lingjuan.lv@sony.com

**Xiaolin Zheng**
College of Computer Science
Zhejiang University
Hangzhou, China
xlzheng@zju.edu.cn

**Dan Meng**
OPPO Research Institute
Shenzhen, China
mengdan90@163.com

**Jun Wang**
OPPO Research Institute
Shenzhen, China
junwang.lu@gmail.com

## Abstract

With growing concerns regarding privacy in machine learning models, regulations have committed to granting individuals the right to be forgotten while mandating companies to develop non-discriminatory machine learning systems, thereby fueling the study of the machine unlearning problem. Our attention is directed toward a practical unlearning scenario, i.e., recommendation unlearning. As the state-of-the-art framework, i.e., RecEraser, naturally achieves full unlearning completeness, our objective is to enhance it in terms of model utility and unlearning efficiency. In this paper, we rethink RecEraser from an ensemble-based perspective and focus on its three potential losses, i.e., redundancy, relevance, and combination. Under the theoretical guidance of the above three losses, we propose a new framework named `UltraRE`, which simplifies and powers RecEraser for recommendation tasks. Specifically, for redundancy loss, we incorporate transport weights in the clustering algorithm to optimize the equilibrium between collaboration and balance while enhancing efficiency; for relevance loss, we ensure that sub-models reach convergence on their respective group data; for combination loss, we simplify the combination estimator without compromising its efficacy. Extensive experiments on three real-world datasets demonstrate the effectiveness of `UltraRE`. The source codes are available at `https://github.com/ZhangYizhao/UltraRE`.

## 1 Introduction

Machine Learning (ML) models have made significant strides in various domains, including natural language processing [36], image recognition [15], and recommender systems [18, 27, 28]. However, privacy concerns arise due to individual data involved in training ML models. These concerns are

---

[*]Corresponding author.

37th Conference on Neural Information Processing Systems (NeurIPS 2023).

mainly twofold. Firstly, the General Data Protection Regulation (GDPR) [44] provides individuals with the right to request the removal of their data, including any impact the data may have on the trained models, i.e., the right to be forgotten. Secondly, the Algorithmic Accountability Act [31] requires companies to evaluate the effect of their ML systems on bias and discrimination.

Machine unlearning is a practical approach that promotes privacy in ML systems by removing previously used data and learned information from ML models. Individuals can remove sensitive information that is learned through their data, while companies can proactively unlearn biased [32] or inaccurate [24] data. Unlearning methods can be divided into two approaches based on the level of completeness, namely exact unlearning (full completeness), and approximate unlearning (partial completeness).

In this paper, we concentrate on recommendation unlearning. The personalized recommendation is a typical scenario that urgently requires unlearning because (i) recommender systems rely heavily on individual data, and (ii) the performance of recommendation is highly sensitive to the quality of training data [40]. Existing recommendation unlearning method [7] follows the Exact Unlearning (EU) approach. As retraining is an algorithmic way to ensure fully complete unlearning, EU approach is mainly developed on the ensemble retraining framework [4, 26, 7, 48]. Similar to the idea of ensemble learning [39], the ensemble retraining framework involves dividing the dataset into non-overlapping shards, training a sub-model for each shard independently, and ultimately combining all sub-models. The ensemble retraining framework limits retraining overhead to sub-models, thereby avoiding retraining from scratch on the entire dataset.

However, the ensemble retraining framework's non-overlapping division isolates user and item collaboration, which leaves considerable room for performance improvement in recommendation tasks. Guided by the intuition of collaboration preservation, RecEraser groups similar data together and combines sub-models with attention networks. In this paper, we analyze RecEraser as an ensemble system through a theoretical lens, and find that i) existing clustering algorithms exhibit an incongruity between the requirements of collaboration and balance, and ii) the usage of attention networks is not necessary. As shown in Figure 1, the lower bound of error rate $\mathcal{B}(\varepsilon)$ in an ensemble system can be decomposed into three components, i.e., redundancy, relevance, and combination loss [34]. Each of these components is associated with a particular stage of the ensemble retraining framework. Considering the three loss components as a well-grounded set of metrics, we simplify and power the State-Of-The-Art (SOTA) recommendation unlearning framework, i.e., RecEraser. Specifically, for redundancy loss (stage I), we incorporate transport weights in the clustering algorithm to pursue the optimal trade-off between collaboration and balance; for relevance loss (stage II), we ensure that sub-models reach convergence on their respective shard data; for combination loss (stage III), we simplify the design of the sub-model combiner to reduce complexity without compromising its efficacy. As a result, we propose a novel recommendation unlearning framework named Ultra RecEraser (`UltraRE`) that offers greater model utility and efficiency while still achieving full completeness. The main contributions of this paper are summarized as follows:

- We propose a novel ensemble retraining framework (`UltraRE`) to address the problem of recommendation unlearning. `UltraRE` enhances both model utility and unlearning efficiency while achieving unlearning completeness at the algorithmic level.

- During stage I (non-overlapping division), we propose an optimal balanced clustering algorithm that transforms the discrete clustering problem into a continuous optimization process while incorporating a balanced constraint to achieve both balanced clustering and minimal inertia simultaneously.

- During stage III (model combination), we take an empirical investigation into the choice of model combiners, and simplify the complexity of model combiner without compromising model utility.

- We empirically validate and demonstrate the proposed framework through extensive experiments on three real-world datasets in terms of model utility and unlearning efficiency.

## 2  Preliminary

In this section, we first briefly introduce the goals and targets in recommendation unlearning, followed by the error decomposition of an ensemble system.

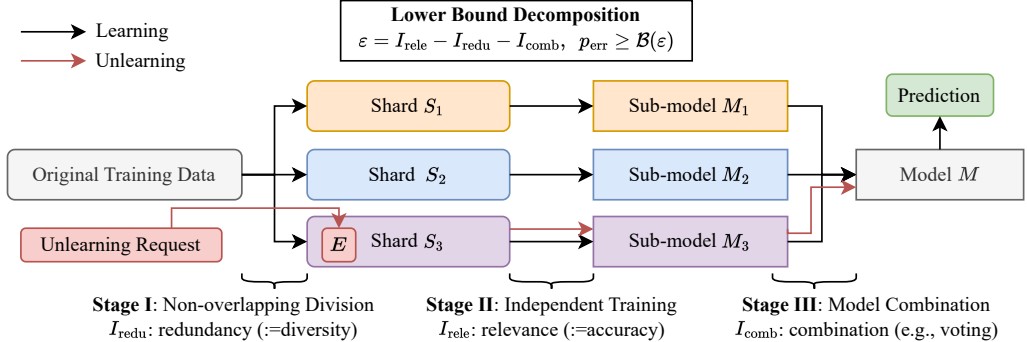

Figure 1: Decomposition of error rate [34] in the ensemble retraining framework which is the SOTA framework for model-agnostic exact unlearning and recommendation unlearning.

## 2.1 Recommendation Unlearning

**Unlearning Goals.** As introduced in [7, 26], there are mainly three goals for an unlearning task: (G1) Unlearning completeness, which requires completely eliminating the impact of target data from a previously trained model; (G2) Unlearning efficiency, which entails maximizing the efficiency of unlearning while minimizing the need for computationally expensive retraining; and (G3) Model utility, which ensures that the unlearned model can achieve recommendation performance comparable to that of the model retrained from scratch.

As the ensemble-retaining framework belongs to the EU approach, which naturally achieves G1 at an algorithmic level, i.e., the highest level of completeness [43]. In this paper, our objective is to enhance the quality of the ensemble-retraining framework within the recommendation context, with a view to advancing G2 and G3.

**Unlearning Targets.** The concept of training data holds a multifaceted view, resulting in different unlearning targets. In the context of personalized recommendation, the user-item interaction serves as the training data. Considering ratings as a form of user-item interaction, the unlearning target may differ based on the perspective, either focusing on the user-wise, item-wise, or sample-wise aspects. For instance, user-wise unlearning refers to unlearning all interactions made by the specific target user(s). In this paper, we focus on the ensemble retraining framework, which is versatile and can be applied to any type of unlearning target.

## 2.2 Ensemble System

Ensemble learning has proven to be successful in various fields of machine learning. An ensemble system involves combing multiple models, such as bagging [5, 23], stacking [46], and mixture of experts [20, 42]. Early studies have been guided by the intuition that ensemble systems achieve better performance when employing a combination of accurate and diverse models. From a theoretical standpoint, using Fano's inequality of information theory, [34] decomposes the error rate of an ensemble system into three components, i.e., redundancy (model diversity), relevance (model accuracy), and combination (information lost during model combination) loss. As shown in Figure 1, the aforementioned ensemble retraining framework is basically an ensemble system. We divide the ensemble retraining into three stages, i.e., non-overlapping division, independent training, and model combination. Each stage is associated with one specific error component.

## 3 Related Work

### 3.1 Machine Unlearning

Machine unlearning is the process of removing the influence of specific training data, i.e., unlearning target, from a learned model [35]. A naive approach to achieve this is by retraining the model from

scratch on the updated dataset that excludes the target. However, this approach can be computationally prohibitive in practice. Based on the degree of unlearning completeness, existing unlearning methods can be classified into the following two approaches.

**Exact Unlearning.** This approach aims to ensure that unlearning target is fully unlearned, i.e., as complete as retraining from scratch. Cao and Yang [6] achieved this by transforming training data points into a reduced number of summations to enhance unlearning efficiency. Recently, Bourtoule et al. [4] proposed an ensemble retraining framework, i.e. SISA, which divides the dataset into non-overlapping subsets, trains a sub-model on each subset, and combines all sub-models in the end. This design reduces the retraining overhead to subsets. Similarly, ARCANE [48] divides the dataset by class and applies one-class anomaly detection training on each subset. However, ARCANE can only be applied to classification tasks.

**Approximate Unlearning.** This approach estimates the influence of unlearning target, and removes the influence through direct parameter manipulation [11, 13, 41, 45]. While theoretically more efficient than exact unlearning, this approach faces challenges regarding unlearning completeness, i.e., exactness, due to the inaccurate estimation of influence. In this approach, the influence of unlearning target is estimated by influence function [21, 22], which is found to be fragile in deep learning [1]. Recent studies also point out that the influence of individual training data on deep models is intractable to compute analytically [12].

## 3.2 Recommendation Unlearning

RecEraser was proposed to achieve unlearning in recommender systems [7]. Following SISA's ensemble retraining framework, RecEraser groups similar data together, instead of random division. This modification effectively preserves collaborative information necessary for personalized recommendations. In addition, RecEraser uses an attention-based combination to further enhance model utility. Similarly, LASER also groups similar data together [26]. However, instead of training a model on each subset and combining them, LASER trains a model sequentially on each subset using curriculum learning. While this modification significantly enhances model utility, it comes at the cost of reduced efficiency. Theoretically, LASER can only accelerate the unlearning speed two times compared to retraining from scratch, which is generally unsatisfying in practice. Recently, Approximate recommendation unlearning was proposed to enhance efficiency [25]. However, it still suffers from common weaknesses of approximate unlearning and is unable to provide algorithmic unlearning completeness, which exact unlearning achieves.

# 4 Methodology

In this section, we present our `UltraRE` for addressing the recommendation unlearning problem. Following the design of the ensemble retraining framework, we rethink RecEraser from an ensemble-based perspective. Instructed by the error decomposition theory in ensemble systems (see Figure 1), we associate each loss component with a specific stage in the ensemble retraining framework. Our proposed `UltraRE` refines the design of each stage to minimize its corresponding loss component. In the following subsections, we describe in detail the modifications we propose for each stage.

## 4.1 Redundancy Loss (Stage I)

In stage I, the ensemble retraining framework divides the original training data into several non-overlapping shards.

**Random Balanced Division.** The original framework (SISA [4], designed for the machine unlearning problem), uses random balanced division. The assumption that there is no prior knowledge regarding the distribution of unlearning requests is widely accepted and practical [4, 7, 26]. In the absence of such knowledge, it is optimal to presume that users submit these requests with equal probability. Therefore, in order to attain optimal unlearning efficiency, the ensemble retaining framework needs to achieve balanced division. The theoretical explanation can be found in Appendix A.

**Balanced $k$-means.** Collecting collaborative information across users and items is essential to the performance of recommendation models. With the intuition of preserving collaboration, RecEraser [7] uses clustering algorithms to group similar samples together. Additionally, RecEraser needs to achieve balanced division. Thus, the balanced clustering algorithms were proposed [7, 8, 26]. Assume that there are $N$ input samples $\boldsymbol{x} \in \mathbb{R}^d$ for clustering. Note that the input samples can consist of user embedding, item embedding, or a combination of both depending on the unlearning target, i.e., user-wise, item-wise, or sample-wise. The core idea of their proposed algorithm is i) limiting the maximum number of samples in one shard, typically $|S_i| \leq \lceil N/k \rceil$, ii) constructing a priority list for every *sample-centroid* pair, and iii) assigning each sample to its destination shard by priority value.

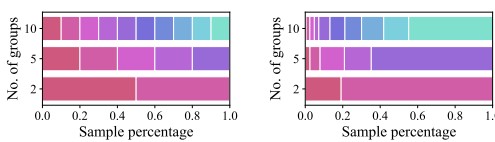

(a) Balanced Grouping    (b) Imbalanced Grouping

Figure 2: An illustration of balanced and imbalanced grouping. We conducted an experiment on a real-world dataset (ML-1M). (a) and (b) are the results of random balanced division and $k$-means respectively. Each color block represents one shard. The size of the block varies with the number of samples in the shard, with larger blocks representing more samples.

### 4.1.1 Optimal Balanced Clustering

The loss component for stage I is redundancy loss, which directs ensemble systems to enhance the diversity among shards [34]. This theoretical guidance conforms with prior work's intuition of collaboration preservation, which proposes the balanced $k$-means algorithm to group similar samples together [7]. However, to achieve a balanced division, balanced $k$-means may not always assign samples to their optimal clusters. In practice, as shown in Figure 3, this can result in a significant number of sub-optimally assigned samples, which degrades the clustering performance. To address this incongruity, we propose an Optimal Balanced Clustering (OBC) algorithm that achieves an adaptive equilibrium between **sample similarity** and **shard balance**.

Generally speaking, OBC incorporates a balanced constraint into the optimization process. As illustrated in Figure 3, we obtain the trans-

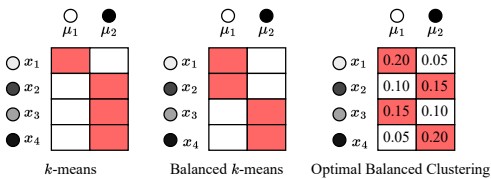

Figure 3: An illustration of different clustering algorithms. The red block located at coordinate $(\boldsymbol{x}_i, \boldsymbol{\mu}_j)$ represents assigning $\boldsymbol{x}_i$ to $\boldsymbol{\mu}_j$. On the left, $k$-means groups samples based solely on their similarity, resulting in an imbalanced result. In the middle, balanced $k$-means forces a balanced result by considering a similarity-based priority list. On the right, Optimal Balanced Clustering (OBC) incorporates a balanced constraint into the optimization process and assigns samples according to their transport weight values.

port weight of assigning an input sample $\boldsymbol{x}$ to a cluster centroid $\boldsymbol{\mu}$ through the optimization process. The input samples are assigned to the cluster with the largest weight. To start with the derivation of OBC, we first introduce the basic concept of $k$-means. The $k$-means algorithm minimizes the total distance of each sample-centroid pair, namely inertia $\mathcal{I}$. It is formally defined as

$$\mathcal{I} = \sum_{j=1}^{k} \sum_{i \in S_j} \|\boldsymbol{x}_i - \boldsymbol{\mu}_j\|_2^2, \quad \boldsymbol{\mu}_j = \frac{\sum_{i \in S_j} \boldsymbol{x}_i}{|S_j|}. \tag{1}$$

Considering $\boldsymbol{x}$ and $\boldsymbol{\mu}$ as two variables respectively sampled from subsets $X$ and $Y$ in the Euclidean space $\mathbb{R}^d$, the problem of inertia minimization can be framed as the Monge-Kantorovich problem.

**Problem 1** (Monge-Kantorovich Problem). *Given the transport cost function $c : X \times Y \to \mathbb{R}$, the objective of the Monge-Kantorovich problem is to find the joint probability measure $P : X \times Y \to \mathbb{R}$ that minimizes the total transport cost*

$$\mathbb{E}[c(X, Y)] = \min_P \int_{X \times Y} c(\boldsymbol{x}, \boldsymbol{\mu}) dP(\boldsymbol{x}, \boldsymbol{\mu}). \tag{2}$$

The Monge-Kantorovich problem offers the advantage of accommodating additional constraints to the probability measure. Turning back to the inertia minimization problem, we insert a set of transport

weights $\boldsymbol{w}$ where their summation equals one into Eq (1) to mimic the function of the joint probability measure $P$. Then, we can rewrite the objective of $k$-means as

$$\min_{\boldsymbol{w}} \left[ \sum_{j=1}^{k} \sum_{i=1}^{N} w_{ij} \cdot \|\boldsymbol{x}_i - \boldsymbol{\mu}_j\|_2^2 \right], \quad s.t. \sum_{j=1}^{k} \sum_{i=1}^{N} w_{ij} = 1, w_{ij} \in \{0, \frac{1}{N}\}, \sum_{i=1}^{N} w_{ij} = \frac{1}{k}, \quad (3)$$

where $w_{ij} = 1/N$ denotes assigning $\boldsymbol{x}_i$ to $\boldsymbol{\mu}_j$, and the constraint $\sum_j w_{ij} = 1/k$ ensures each shard is treated equally. By introducing the transport weights $\boldsymbol{w}$, we can transform the discrete clustering problem into a continuous optimization process. This transformation allows us to impose additional constraints on $\boldsymbol{w}$ that facilitate more fine-grained control over the clustering process. Consequently, we add the constraint $\sum_j w_{ij} = 1/N$ to ensure a balanced division. Following the Monge-Kantorovich framework [3, 30], we also relax the range of $\boldsymbol{w}$ to $\mathbb{R}^+$ to guarantee the existence of solution space for the objective. Finally, the shard assignment of $\boldsymbol{x}_i$ is determined by $\arg\max_j w_{ij}$. However, the worst-case complexity of computing the optimum for such a transport objective with additional constraints scales in $O(N^3)$ [37]. To enhance efficiency, we utilize Sinkhorn divergence to accelerate the optimization process [38, 29]. Specifically, we smooth the objective with an entropic regularization as follows:

$$\min_{\boldsymbol{w} \in \Gamma} \left[ \sum_{j=1}^{k} \sum_{i=1}^{N} w_{ij} \cdot \|\boldsymbol{x}_i - \boldsymbol{\mu}_j\|_2^2 + \epsilon \cdot \sum_{j=1}^{k} \sum_{i=1}^{N} w_{ij} \cdot (\log(w_{ij}) - 1) \right], \quad (4)$$

where $\Gamma = \{\|\boldsymbol{w}\|_1 = 1, \boldsymbol{w} \succeq 0, \sum_i w_{ij} = \frac{1}{k}, \sum_j w_{ij} = \frac{1}{N}\}$. The derived new objective can be efficiently solved through Sinkhorn's matrix scaling algorithm with a complexity of $O(Nk)$ [9, 10]. The optimization details can be found in Appendix B.

## 4.2 Relevance Loss (Stage II)

In stage II, the ensemble retraining framework trains a sub-model on each shard independently, which means sub-models do not interfere with each other during training. The loss component associated with this stage is relevance loss, which implies that the aim is to enhance the performance of sub-models. However, to ensure unlearning completeness at the algorithmic level, it is crucial that the sub-models fully replicate the original model. This includes replicating not only the model structure, but also hyper-parameters, parameter initialization, and any other relevant elements. Therefore, we do not break the requirement of full replication and also leave stage II unchanged [4, 7, 8]. We assume that the shard data is i.i.d. with the original training data and ensure that all sub-models fully replicate the original model that attains convergence on the original training data. This ensures that sub-models reach convergence on their respective shard data.

## 4.3 Combination Loss (Stage III)

In stage III, the ensemble retaining framework combines the sub-models to obtain the final model. The original framework, i.e., SISA [4], uses an average combiner. However, this is a naive solution, since various shards may have different contributions to the final model. Thus, the model-based combiner is proposed by [7, 8]. Specifically, this approach utilizes machine learning models to determine the combination weights by the following objective:

$$\min_{\boldsymbol{\beta}} \mathbb{E} \left[ \sum_{i=1}^{k} \ell(\beta_i \boldsymbol{\alpha_i}, R) \right] + \lambda \|\boldsymbol{\beta}\|_2^2, \quad s.t. \ \|\boldsymbol{\beta}\|_1 = 1, \boldsymbol{\beta} \succeq 0, \quad (5)$$

where $\ell$ is the original loss function for recommendation tasks, $R$ is the recommendation data, $\boldsymbol{\alpha_i}$ denotes the parameters of $i$-th sub-models, and $\boldsymbol{\beta}$ denotes the weights of combination. Note that the parameters of sub-models are fixed during model combination. The $L_2$ regularization parameterized by $\lambda$ on $\boldsymbol{\beta}$ is applied to prevent over-fitting. This approach is akin to the use of meta-estimators in ensemble systems [33], where they are employed to alleviate information loss during model combination. As empirically studied by [34], a simple meta-estimator such as Logistic Regression (LR) proves to be enough on deep models. Building on this insight, we empirically validate the effectiveness of LR on the recommendation tasks in Section 5.2.3. Our results show that LR can perform comparably to attention networks [7] while substantially reducing model complexity.

## 4.4 Putting Together

Our proposed `UltraRE` belongs to the ensemble retraining framework and follows the three-stage structure described in Figure 1. In stage I, we obtain an adaptive equilibrium between sample similarity and shard balance by optimizing Eq (4) In stage II, we do not interfere with sub-model training, meeting the requirement of fully replicating the original model. In stage III, we apply LR to determine the combination weight by optimizing Eq (5).

# 5 Experiments

We conduct experiments on three real-world datasets to evaluate the performance of our proposed `UltraRE` framework. The evaluation mainly focuses on G2 (unlearning efficiency) and G3 (model utility), as our proposed method (`UltraRE`) belongs to exact unlearning, which naturally achieves G1 (unlearning completeness). `UltraRE` can handle all three types of unlearning targets, i.e., user-wise, item-wise, and sample-wise. In this paper, we focus on user-wise unlearning without loss of generality, as it is the most common one in practice. Additionally, we perform an ablation study to further investigate the effectiveness of modification.

## 5.1 Experimental Settings

### 5.1.1 Datasets

We conduct experiments on the following three real-world datasets: i) MovieLens 100k (**ML-100K**)[2]: The MovieLens datasets are among the most extensively used in recommendation researchn [14, 16]. ML-100K contains 100 thousand ratings; ii)

Table 1: Summary of datasets.

| Dataset | User # | Item # | Rating # | Sparsity |
|---------|--------|--------|----------|----------|
| ML-100K | 943 | 1,682 | 100,000 | 93.695% |
| ML-1M | 6040 | 3,950 | 1,000,209 | 95.814% |
| ADM | 478,235 | 266,414 | 836,006 | 99.999% |

MovieLens 1M (**ML-1M**): This is a stable version of the MovieLens dataset, containing 1 million ratings; and iii) Amazon Digital Music (**ADM**)[3]: The Amazon dataset contains several sub-datasets according to the categories of Amazon products. ADM is the sub-dataset containing digital music reviews. To ensure reliable evaluations, we filter out the users and items that have less than 5 interactions. Specifically, we use 80% of ratings for training, 10% as a validation set for tuning hyper-parameters, and the remainder for testing. Table 1 summarizes the statistics of three datasets.

### 5.1.2 Compared Models and Methods

**Recommendation Models.** Our proposed `UltraRE` is model-agnostic, enabling its application to any recommendation model. In this paper, we select two representative recommendation models: i) a classic model, i.e., Deep Matrix Factorization (DMF) [47], and ii) the SOTA model, i.e. Light-GCN [18], for testing. Following the original papers, we adopt normalized binary cross entropy loss and Bayesian personalized ranking loss for DMF and LightGCN respectively, and employ Adam optimizer to train the above models. We run all experiments for 10 trials and report the average results. Following [7], we use WMF [19] as a pre-training model to generate user and item embedding for the purpose of clustering.

**Unlearning Methods.** We compare `UltraRE` with the benchmark and the SOTA methods, including: i) **Retrain**: Retraining the model from scratch on the updated dataset; ii) **SISA** [4]: the SOTA generic exact unlearning method which is based on ensemble retraining framework; and iii) **RecEraser** [7]: the SOTA recommendation unlearning method which modifies SISA to boost performance in recommendation tasks. Following [4, 7], we set the number of shards to 10 for all unlearning methods that involve division, i.e., SISA, RecEraser, and `UltraRE`.

---

[2]https://grouplens.org/datasets/movielens/
[3]http://jmcauley.ucsd.edu/data/amazon/

Table 2: Running time (s) of the learning process.

| ML-100K | DMF | | | | LightGCN | | | |
| --- | --- | --- | --- | --- | --- | --- | --- | --- |
| | Retrain | SISA | RecEraser | UltraRE | Retrain | SISA | RecEraser | UltraRE |
| Stage I | 0.000 | 0.013 | 0.662 | 0.048 | 0.000 | 0.014 | 0.645 | 0.050 |
| Stage II | 1,442.263 | 193.317 | 189.742 | 190.289 | 4,422.531 | 582.659 | 594.714 | 591.638 |
| Stage III | 0.000 | 0.001 | 42.673 | 31.540 | 0.000 | 0.003 | 383.132 | 315.822 |
| Total | 1,442.263 | 193.331 | 233.077 | 221.877 | 4,422.531 | 582.676 | 978.491 | 907.510 |

| ML-1M | DMF | | | | LightGCN | | | |
| --- | --- | --- | --- | --- | --- | --- | --- | --- |
| | Retrain | SISA | RecEraser | UltraRE | Retrain | SISA | RecEraser | UltraRE |
| Stage I | 0.000 | 0.046 | 23.904 | 0.429 | 0.000 | 0.058 | 23.950 | 0.467 |
| Stage II | 3,732.366 | 376.237 | 378.475 | 374.999 | 11,372.446 | 1,218.335 | 1,211.275 | 1,218.875 |
| Stage III | 0.000 | 0.003 | 93.519 | 69.327 | 0.000 | 0.004 | 682.182 | 624.349 |
| Total | 3,732.366 | 376.286 | 495.898 | 444.755 | 11,372.446 | 1,218.397 | 1,917.407 | 1,843.691 |

| ADM | DMF | | | | LightGCN | | | |
| --- | --- | --- | --- | --- | --- | --- | --- | --- |
| | Retrain | SISA | RecEraser | UltraRE | Retrain | SISA | RecEraser | UltraRE |
| Stage I | 0.000 | 0.052 | 20.864 | 0.373 | 0.000 | 0.054 | 20.851 | 0.352 |
| Stage II | 2,012.655 | 207.457 | 201.769 | 206.417 | 6,724.734 | 698.737 | 679.051 | 682.742 |
| Stage III | 0.000 | 0.003 | 57.352 | 48.975 | 0.000 | 0.004 | 404.110 | 374.253 |
| Total | 2,012.655 | 207.512 | 279.985 | 255.765 | 6,724.743 | 698.795 | 1,104.012 | 1,057.347 |

## 5.2 Results and Discussion

### 5.2.1 Unlearning Efficiency (G2)

We use running time to evaluate the efficiency of unlearning. To fully exploit the efficiency of the ensemble retraining framework, we run all shards in parallel. Specifically, we measure the running time of each stage during the learning process and report the results in Table 2. As the shard division (stage I) and combination weights (stage III) are determined during learning and remain unchanged during unlearning, their time cost during unlearning is negligible during unlearning compared to retraining (stage II). Since all shards run in parallel, the running time of stage II remains consistent between learning and unlearning. Therefore, by measuring the running time of stage II during learning, we can evaluate the running time of unlearning. We run all experiments on the same Ubuntu 20.04 LTS System server with 48-core CPU, 256GB RAM and NVIDIA GeForce RTX 3090 GPU. From Table 2, we observe that i) compared with Retrain, the ensemble retraining frameworks, i.e., SISA, RecEraser, and UltraRE, significantly enhance unlearning efficiency. Although spending more time in stages I and III, these frameworks decrease the total running time by an average of 85.39%; ii) Using random balanced division and average combiner, SISA enjoys a notably faster speed than other ensemble retraining frameworks. Nevertheless, this simple design also limits SISA's performance regarding model utility (see Section 5.2.2); and iii) Among the comparison of recommendation unlearning frameworks, i.e., RecEraser and UltraRE, our proposed UltraRE decreases the running time of both stages I and III. In stage I, our proposed optimal balanced clustering algorithm can improve the efficiency by 98.11% on average. This improvement is even greater when dealing with larger datasets, i.e., ML-1M and ADM. In stage III, we simplify the choice of the model combiner, resulting in an average efficiency increase of 11.95%.

### 5.2.2 Model Utility (G3)

We use two common metrics, i.e., Normalized Discounted Cumulative Gain (NDCG) and Hit Ratio (HR), to evaluate the performance of recommender models [17, 47]. For both metrics, we truncate the ranked list at 10, and report NDCG@10 and HR@10 during both the learning and unlearning processes. To simulate user-wise unlearning, we randomly select $q\%$ of users to unlearn, where $q$ is investigated in {5, 10} across all datasets. From Table 3, we observe that i) Retrain achieves the best performance, suggesting using ensemble retraining frameworks (including SISA, RecEraser, and UltraRE) may come at the cost of slightly increased error rates. This trade-off occurs because these frameworks prioritize unlearning efficiency over preserving model utility; ii) The performance of cluster-based division methods, i.e., RecEraser and UltraRE, surpasses that of the random division method, i.e., SISA. This implies that by clustering similar samples together, redundancy loss can be

Table 3: Recommendation performance during learning and unlearning (G3: model utility). The best results except Retrain are highlighted in **bold**. The superscript ∗ indicates $p < 0.01$ for the t-test of `UltraRE` against RecEraser.

| ML-100K | | DMF | | | | LightGCN | | | |
|---|---|---|---|---|---|---|---|---|---|
| | | Retrain | SISA | RecEraser | UltraRE | Retrain | SISA | RecEraser | UltraRE |
| Learn | NDCG | 0.3956 | 0.3716 | 0.3795 | **0.3847*** | 0.3997 | 0.3684 | 0.3812 | **0.3859*** |
| | HR | 0.4374 | 0.4227 | 0.4273 | **0.4326*** | 0.4395 | 0.4203 | 0.4257 | **0.4312*** |
| Unlearn@5 | NDCG | 0.3934 | 0.3709 | 0.3762 | **0.3815*** | 0.3976 | 0.3677 | 0.3799 | **0.3841*** |
| | HR | 0.4359 | 0.4222 | 0.4232 | **0.4297*** | 0.4387 | 0.4201 | 0.4233 | **0.4295*** |
| Unlearn@10 | NDCG | 0.3905 | 0.3702 | 0.3734 | **0.3786*** | 0.3953 | 0.3669 | 0.3793 | **0.3836*** |
| | HR | 0.4353 | 0.4205 | 0.4183 | **0.4245*** | 0.4362 | 0.4186 | 0.4221 | **0.4267*** |

| ML-1M | | DMF | | | | LightGCN | | | |
|---|---|---|---|---|---|---|---|---|---|
| | | Retrain | SISA | RecEraser | UltraRE | Retrain | SISA | RecEraser | UltraRE |
| Learn | NDCG | 0.4382 | 0.3956 | 0.3973 | **0.4042*** | 0.4437 | 0.3941 | 0.4171 | **0.4241*** |
| | HR | 0.5402 | 0.5141 | 0.5134 | **0.5183*** | 0.5493 | 0.4672 | 0.5160 | **0.5217*** |
| Unlearn@5 | NDCG | 0.4398 | 0.3937 | 0.3965 | **0.4031*** | 0.4403 | 0.3908 | 0.4138 | **0.4189*** |
| | HR | 0.5411 | 0.5125 | 0.5129 | **0.5179*** | 0.5464 | 0.4543 | 0.5140 | **0.5206*** |
| Unlearn@10 | NDCG | 0.4391 | 0.3914 | 0.3963 | **0.4013*** | 0.4396 | 0.3912 | 0.4129 | **0.4181*** |
| | HR | 0.5411 | 0.5108 | 0.5127 | **0.5184*** | 0.5421 | 0.4495 | 0.5090 | **0.5146*** |

| ADM | | DMF | | | | LightGCN | | | |
|---|---|---|---|---|---|---|---|---|---|
| | | Retrain | SISA | RecEraser | UltraRE | Retrain | SISA | RecEraser | UltraRE |
| Learn | NDCG | 0.4549 | 0.4063 | 0.4234 | **0.4294*** | 0.4603 | 0.4025 | 0.4281 | **0.4345*** |
| | HR | 0.7914 | 0.7012 | 0.7230 | **0.7311*** | 0.7931 | 0.6912 | 0.7406 | **0.7472*** |
| Unlearn@5 | NDCG | 0.4551 | 0.4021 | 0.4214 | **0.4273*** | 0.4594 | 0.3963 | 0.4251 | **0.4311*** |
| | HR | 0.7915 | 0.6959 | 0.7202 | **0.7283*** | 0.7945 | 0.6851 | 0.7311 | **0.7403*** |
| Unlearn@10 | NDCG | 0.4554 | 0.3973 | 0.4198 | **0.4262*** | 0.4595 | 0.4043 | 0.4239 | **0.4306*** |
| | HR | 0.7916 | 0.6772 | 0.7222 | **0.7298*** | 0.7936 | 0.6781 | 0.7192 | **0.7255*** |

significantly reduced. iii) Our proposed `UltraRE` demonstrates superior performance compared to the SOTA recommendation unlearning framework, i.e., RecEraser. On average, our model achieved an improvement in recommendation metrics of 1.31%, 1.26%, and 1.26%, on ML-100k, ML-1M and ADM respectively. The improved results can be attributed to `UltraRE`'s ability to generate better clustering outcomes while meeting the balance requirement; and iv) Strong performance during the learning process serves as a reliable indicator of strong performance during unlearning.

### 5.2.3 Ablation Study

To fully understand the effectiveness of our proposed `UltraRE`, we conduct ablation studies regarding the main modifications during stages I (non-overlapping division) and III (model combination), as well as varying numbers of shards.

**Effect of Division.** In stage I, we propose a novel division algorithm, named Optimal Balanced Clustering (OBC). We conduct an ablation study to compare OBC with Balanced Random Division (BRD, used in SISA [4]), Balanced $k$-means (BKM, used in RecEraser [7]) and $k$-means (KM). Although KM causes the issue of imbalance in clustering, it can still be a valuable baseline for evaluating division performance. We

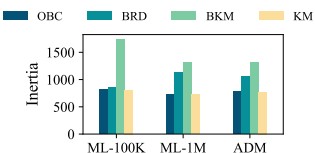

Figure 4: Effect of division (Stage I)

use inertia in Eq (1) to evaluate division performance. A lower inertia value indicates better division performance. We present the results obtained with $k = 10$ in Figure 4. From it, we observe that i) Among balanced grouping algorithms, our proposed OBC significantly outperforms BKM (by 46.32% on average) and BRD (by 22.01% on average); ii) OBC also achieves comparable inertia to KM, indicating that OBC achieves balanced grouping without compromising sample similarity, and iii) BRD even outperforms BKM. This indicates that the usage of a priority list in BKM leads to a significant degradation in division performance, making it worse than random division.

**Effect of Combination.** In stage III, we simplify the choice of the model combiner. To validate our choice, we compare our logistic regression (LR) based combiner with Average combiner (AVG which is used in SISA [4]) and Attention combiner (ATT which is used in RecEraser [7]). As shown in Figure 5, model-based combiners, i.e., LR and ATT, outperform AVG by a significant margin. On average, AVG's performance is decreased by 3.53% on DMF and

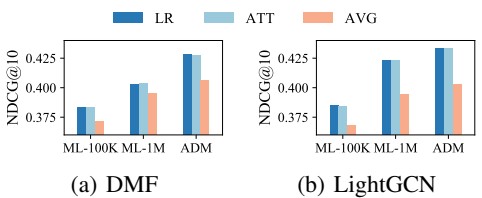

Figure 5: Effect of combination (Stage III).

6.51% on LightGCN. Among model-based combiners, LR can achieve comparable performance with ATT while substantially reducing the model complexity.

**Effect of Shard Number.** To demonstrate the robustness of `UltraRE`, we empirically study the effect of shard number and compare it with the SOTA recommendation unlearning framework RecEraser. We investigate the number of shards $S$ in $\{10, 20, 50\}$. As shown in Figure 6, `UltraRE` yields consistent improvements over RecEraser regarding both unlearning efficiency (time) and model utility (NDCG@10). Moreover, as the shard number increases, `UltraRE`

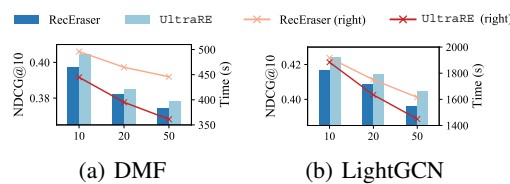

Figure 6: Effect of shard number (ML-1M).

exhibits greater efficiency enhancement over RecEraser. This is because a larger shard number reduces the time taken in stage II, thereby highlighting the efficiency of `UltraRE` in stages I and III.

## 6 Conclusion

In this paper, we refine the SOTA exact unlearning framework, i.e., ensemble retraining framework, in recommendation tasks. Our proposed `UltraRE` breaks away from the prior work's intuition on preserving collaboration [7, 8, 26]. Instead, it is guided by the theoretical analysis of error decomposition in ensemble systems. Among the three stages in the ensemble retraining framework, our modifications mainly lie in stages I (non-overlapping division) and III (model combination). In stage I, we proposed an optimal balanced clustering algorithm that can achieve an adaptive equilibrium between sample similarity and shard balance. In stage III, we simplify the complexity of the model combiner without increasing the combination loss. Extensive experiments on three real-world recommendation datasets demonstrate that `UltraRE` can not only greatly enhance unlearning efficiency, but also outperform the SOTA unlearning models in terms of model utility.

## 7 Broader Impacts and Limitations

Recommendation unlearning can have various implications on society, including addressing issues related to privacy, fairness, bias, and manipulation. `UltraRE` can also be generalized to other unlearning tasks and is especially adept in association-sensitive tasks, e.g., graph learning. A common limitation of the ensemble retraining framework is the absence of experiments on large-scale datasets with a large shard number [4, 7, 8, 26]. As shown in Section 5.2.1, the major time of unlearning is spent on stage II, which cannot fully demonstrate the efficiency of the ensemble retraining framework. We manage to investigate the effect of large shard number in Section 5.2.3.

## Acknowledgments and Disclosure of Funding

This work was supported in part by Leading Expert of Ten Thousands Talent Program of Zhejiang Province (No.2021R52001) and the National Natural Science Foundation of China (No.62172362).

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

# A    Theoretical Validation of Balanced Division

Supposing there is no prior knowledge regarding the distribution of unlearning requests, it is optimal to presume that users submit requests with equal probability. Under this condition, a balanced division in the ensemble retraining framework can achieve maximum unlearning efficiency.

*Proof.* Let $h_i$ denote the training overhead of shard $S_i$, which is in proportion to the shard size, i.e., the number of samples in the shard. Since the users submit unlearning requests with equal probability, the probability of a request located in shard $S_i$ is $P_i = h_i/Z$ where $Z = \sum_j h_j$. Thus, the expectation of retraining overhead when dealing with an unlearning request is

$$\mathbb{E}(H) = \sum_{i=1}^{k} P_i \cdot h_i = \frac{1}{Z} \sum_{i=1}^{k} h_i^2. \tag{6}$$

According to Cauchy-Schwarz Inequality [2], for $n$ random variables $x_i$, we can get a lower bound of $\sum_i x_i^2$ as:

$$\sum_{i=1}^{n} x_i^2 \geq \frac{(\sum_{i=1}^{n} x_i)^2}{n}. \tag{7}$$

Taking it into (6), we have

$$\mathbb{E}(H) = \frac{1}{Z} \sum_{i=1}^{k} h_i^2 \geq \frac{Z}{k}. \tag{8}$$

We can easily find that setting $h_i = Z/k$ achieves this lower bound, which indicates that a balanced division achieves the maximum unlearning efficiency. □

# B    Sinkhorn Algorithm

In this section, we provide the details of optimizing the objective of the optimal balanced clustering algorithm. Firstly, we smooth the objective with an entropic regularization term:

$$\min_{\boldsymbol{w} \in \Gamma} \left[ \sum_{j=1}^{k} \sum_{i=1}^{N} w_{ij} \cdot \|\boldsymbol{x}_i - \boldsymbol{\mu}_j\|_2^2 + \epsilon \cdot \sum_{j=1}^{k} \sum_{i=1}^{N} w_{ij} \cdot (\log(w_{ij}) - 1) \right]. \tag{9}$$
$$s.t. \quad \|\boldsymbol{w}\|_1 = 1, \boldsymbol{w} \succeq 0, \sum_i w_{ij} = \frac{1}{k}, \sum_j w_{ij} = \frac{1}{N}$$

We rewrite Eq (9) with Lagrange multipliers as

$$\max_{\boldsymbol{f}, \boldsymbol{g}} \min_{\boldsymbol{w}} \mathcal{J} = \left\{ \sum_{j=1}^{k} \sum_{i=1}^{N} w_{ij} c_{ij} + \epsilon \cdot \sum_{j=1}^{k} \sum_{i=1}^{N} w_{ij} \cdot (\log(w_{ij}) - 1) - \sum_{j=1}^{k} f_j \left[ \left( \sum_{i=1}^{N} w_{ij} \right) - \frac{1}{k} \right] \right.$$
$$\left. - \sum_{i=1}^{N} g_i \left[ \left( \sum_{j=1}^{k} w_{ij} \right) - \frac{1}{N} \right] \right\}, \quad c_{ij} = \|\boldsymbol{x_i} - \boldsymbol{\mu}_j\|_2^2. \tag{10}$$

Taking the differentiation w.r.t. $w_{ij}$ on Eq (10), we have

$$\frac{\partial \mathcal{J}}{\partial w_{ij}} = 0 \quad \Rightarrow \quad c_{ij} + \epsilon \log(w_{ij}) - f_j - g_i = 0. \tag{11}$$

To update our variables, we first fix $g_i$ and update $f_j$ with

$$f_j^{(t+1)} = \epsilon \left\{ \log \left( \frac{1}{k} \right) - \log \left[ \sum_{i=1}^{N} \exp \left( \frac{g_i^{(t)} - c_{ij}}{\epsilon} \right) \right] \right\}. \tag{12}$$

Then we fix $f_j$ and update $g_i$ with

$$g_i^{(t+1)} = \epsilon \left\{ \log \left( \frac{1}{N} \right) - \log \left[ \sum_{j=1}^{k} \exp \left( \frac{f_j^{(t)} - c_{ij}}{\epsilon} \right) \right] \right\}. \tag{13}$$

In summary, we can iteratively update $f_j$ and $g_i$ until we obtain the final solutions.

Table 4: Running time of the learning process on ML-10M.

| ML-10M | DMF | | LightGCN | |
|---|---|---|---|---|
| | RecEraser | UltraRE | RecEraser | UltraRE |
| Stage I | 872.53m | 259.18s | 879.06m | 257.83s |
| Stage II | 213.55s | 208.44s | 860.12s | 852.32s |
| Stage III | 83.74s | 67.26s | 384.50s | 376.57s |
| Total | 877.48m | 534.88s | 899.80m | 1,486.72s |

## C  Additional Experiments on the Large-scale dataset

As illustrated in Section 7, current experiments do not sufficiently demonstrate the efficiency enhancement of UltraRE, because the majority of time is spent during stage II (independent training) while the enhancements occur in stages I (non-overlapping division) and III (model combination). Thus, following [7], we further conduct experiments on ML-10M (the largest dataset used in [7]) with 50 shards (a large shard number), and report the results in Table 4. From it, we observe that UltraRE significantly improves efficiency compared to RecEraser. Specifically, in stages I and III, UltraRE demonstrates average efficiency enhancements of 20,227.87% and 13.30% respectively. Note that, in the large-scale dataset, our proposed clustering algorithm (OBC) outperforms the BKM used in RecEraser, reducing clustering time from several hours to just a few minutes (in stage I, 872.53m/879.06m vs 259.18s/257.83s). The experimental results demonstrate higher efficiency improvements when compared to the results reported in the main text, providing additional evidence of the substantial efficiency enhancements achieved by our method.

