# OpenReview forum: "UltraRE: Enhancing RecEraser for Recommendation Unlearning via Error Decomposition"
_NeurIPS.cc/2023/Conference — NeurIPS 2023 poster_

### Official Review · Reviewer_Jkju · 2023-06-28

**Soundness:** 3 good
**Presentation:** 3 good
**Contribution:** 3 good
**Rating:** 7
**Confidence:** 5

**Summary:**

This paper studies the recommendation unlearning problem and focuses on the exact unlearning approach. The authors rethink the exact unlearning framework from an ensemble-based perspective, and decompose the error into three components. The authors mainly modify the existing SOTA framework regarding the first and the third components. For the first component, a new optimal balanced clustering algorithm is proposed to improve efficacy and efficiency. For the third component, the model aggregator is simplified to improve efficiency. Extensive experiments are conducted on three real-world datasets and across different recommendation models.

**Strengths:**

1. The topic (recommendation unlearning) is significant and timely. Recommendation is a typical scenario of machine unlearning where the data naturally comes from diverse users.

2. This paper rethinks the exact unlearning framework from an ensemble-based perspective, and modifies the framework with theoretical guidance, which distinguish this work from prior intuition-motivated modification.

3. One key technical contribution of this paper is the proposed optimal balanced clustering algorithm, which incorporates the balanced constraint into the optimization process, addressing the incongruity between user similarity and shard balancing.

4. Extensive experiments are conducted to evaluate the proposed framework. The evaluations include the performance of two unlearning goals and an ablation study of each stage.

5. In general, this paper is well-organized and easy to follow.

**Weaknesses:**

1. The optimization process of Eq (5) is not clear. Is it identical to prior work [6, 7]? Please provide more details.

2. As shown in Table 2, the improvements in stages I and III appear to be significant. But the time costs of these stages are not comparable with that of stage II, where the authors remain unchanged. Therefore, this makes the overall improvement insignificant.

3. There are some typos in the paper:
Line 203 “\sum_i w_{ij}” -> “\sum_j w_{ij}”
Line 365 “[6, 7, 24] Instead” -> “[6, 7, 24]. Instead”

**Questions:**

Please refer to Weakness 1.

This paper focuses on exact unlearning, but approximate unlearning has gained much attention due to its efficiency. There is also an approximate unlearning method in recommendation unlearning. Although it adopts a different approach, it would be helpful to have a discussion about it.

[a] Li, Y., Chen, C., Zheng, X., Zhang, Y., Gong, B., & Wang, J. (2023). Selective and Collaborative Influence Function for Efficient Recommendation Unlearning. arXiv preprint arXiv:2304.10199.

---

> ### Author Rebuttal · Authors · 2023-08-09
>
> **Q1**:	Optimization process in stage III (model combination).
>
> **Response**: Although different from prior work [6, 7] in terms of the model used, our method also utilizes the stochastic gradient descent algorithm for optimization. We will provide further explanations regarding these details during the revision.
>
> **Q2**:	Significance of efficiency enhancement.
>
> **Response**: As mentioned in the “Broader Impacts and Limitations” section, a common limitation of ensemble retraining frameworks is the lack of experiments on large-scale datasets. Current experiments do not sufficiently demonstrate the efficiency enhancement, because the majority of time is spent during stage II （independent training）while the enhancements occur in stages I (non-overlapping division) and III (model combination). Thus, following [6], we further conduct experiments on ML-10M (large-scale dataset) with 50 shards (large shard number), and report the results in the table below.
>
> | ML-10M    | DMF-RecEraser | DMF-UltraRE | LightGCN-RecEraser | LightGCN-UltraRE |
> |-----------|---------------|-------------|--------------------|------------------|
> | Stage I   | 872.53m       | 259.18s     | 879.06m            | 257.83s          |
> | Stage II  | 213.55s       | 208.44s     | 860.12s            | 852.32s          |
> | Stage III | 83.74s        | 67.26s      | 384.50s            | 376.57s          |
> | Total     | 877.48m       | 534.88s     | 899.80m            | 1,486.72s        |
>
> The results indicate that our proposed UltraRE significantly improves efficiency compared to RecEraser. Specifically, in stages I and III, UltraRE demonstrates average efficiency improvements of **20,227.87%** and 13.30% respectively. Note that, in the large-scale dataset (ML-10M), our proposed clustering algorithm (OBC) significantly outperforms the BKM used in RecEraser, *reducing clustering time from several hours to just a few minutes* (in stage I, 872.53m/879.06m vs 259.18s/257.83s). The experimental results demonstrate higher efficiency improvements when compared to the results reported in the paper, providing additional evidence of the substantial efficiency enhancements achieved by our method.
>
> **Q3**:	Typos.
>
> **Response**: We will carefully review the paper and fix any typos during revision.
>
> **Q4**:	Discussion about approximate recommendation unlearning.
>
> **Response**: We will provide a brief introduction to the mentioned paper in the “Related Work” section.

---

> > ### Comment · Reviewer_Jkju · 2023-08-20
> >
> > The author has addressed all my previous concerns. I will keep my positive rating.

---

### Official Review · Reviewer_55Fw · 2023-07-02

**Soundness:** 3 good
**Presentation:** 3 good
**Contribution:** 3 good
**Rating:** 6
**Confidence:** 4

**Summary:**

This paper tackles the issue in recommendation unlearning algorithm, specifically focusing on RecEraser. The authors introduce UltraRE, a framework devised to optimize the RecEraser. UltraRE aims at mitigating three primary losses - redundancy, relevance, and combination. By integrating transport weights in the clustering algorithm, it addresses redundancy loss, balancing collaboration and balance. Besides, it simplify the complexity of the model combiner without diminishing efficacy. The authors put these modifications together to enhance both unlearning efficiency and model utility, and empirically validate the proposed framework through extensive experiments on three real-world datasets.


**Strengths:**

*  The paper is written in a clear and engaging style, making it easy to follow.
*  It addresses a timely and critical issue, machine unlearning in recommender systems, which is essential in the context of data privacy regulations.
*  The proposed UltraRE framework is straightforward and effective. And the authors conduct extensive experiments on three real-world datasets to demonstrate the efficacy of the proposed framework.


**Weaknesses:**

* The improvements in efficiency and effectiveness brought by UltraRE are very marginal, raising questions about its practical impact and value.
* The novelty in UltraRE is limited, as it mainly employs the existing method to improve the previous method RecEraser.


**Questions:**

In the ablation study, the authors investigated the inertia performance of clustering algorithms in data partitioning for different algorithms. From Figure 4, it is evident that OBC (UltraRE) and BRD (SISA) significantly outperform BKM (RecEraser) in terms of clustering inertia. However, UltraRE does not show a substantial improvement over RecEraser in unlearning performance, and interestingly, SISA performs worse than RecEraser.
* Could the authors elaborate on the relationship between clustering performance (inertia) and unlearning performance?
* Additionally, what specifically contributes to the improvements presented by the proposed method, UltraRE?

**Limitations:**

yes

---

> ### Author Rebuttal · Authors · 2023-08-09
>
> **Q1**:	Novelty of our proposed UltraRE.
>
> **Response**: Please refer to *Response to Q1* in the rebuttal to all reviewers.
>
> **Q2**:	Relationship between clustering performance (inertia) and unlearning performance.
>
> **Response**: Our proposed UltraRE, being an exact unlearning approach, inherently achieves optimal unlearning performance (completeness). Thus, considering this observation and the comments provided, we infer that the unlearning performance you mentioned refers to model utility (recommendation performance).
>
> Inertia is the summation of the inner-cluster distance of all sample-centroid pairs, ranging in $[0, \infty]$. Model utility is evaluated by NDCG and HR, both of which range in $[0, 1]$. Therefore, the scales of inertia and model utility are different, leading to inconsistent improvements between them. To subtly investigate the relationship between inertia and model utility (NDCG@10), we conduct another ablation study. Specifically, we vary the clustering algorithms in stage I and use Logistic Regression (LR) in stage III for all compared methods, and report corresponding inertia and NDCG (during learning). We choose LR due to its comparable performance to ATTention networks (ATT), as shown in the "Effect of Combination" section. For easy comparison, we also report the result of original RecEraser and SISA.
>
> | ML-100K      | Stage I | Stage III | Inertia (stage I) | DMF-NDCG (stage III) | LightGCN-NDCG (stage III) |
> |--------------|---------|-----------|-------------------|----------------------|---------------------------|
> | UltraRE      | OBC     | LR        | 816.32            | 0.3847               | 0.3859                    |
> | RecEraser-LR | BKM     | LR        | 1,738.24          | 0.3792               | 0.3808                    |
> | RecEraser    | BKM     | ATT       | 1,738.24          | 0.3795               | 0.3812                    |
> | SISA-LR      | BRD     | LR        | 852.50            | 0.3824               | 0.3836                    |
> | SISA         | BRD     | AVG       | 852.50            | 0.3716               | 0.3684                    |
>
> | ML-1M        | Stage I | Stage III | Inertia (stage I) | DMF-NDCG (stage III) | LightGCN-NDCG (stage III) |
> |--------------|---------|-----------|-------------------|----------------------|---------------------------|
> | UltraRE      | OBC     | LR        | 729.46            | 0.4042               | 0.4241                    |
> | RecEraser-LR | BKM     | LR        | 1,318.31          | 0.3968               | 0.4169                    |
> | RecEraser    | BKM     | ATT       | 1,318.31          | 0.3973               | 0.4171                    |
> | SISA-LR      | BRD     | LR        | 1,129.72          | 0.3972               | 0.4173                    |
> | SISA         | BRD     | AVG       | 1,129.72          | 0.3956               | 0.3941                    |
>
> | ADM          | Stage I | Stage III | Inertia (stage I) | DMF-NDCG (stage III) | LightGCN-NDCG (stage III) |
> |--------------|---------|-----------|-------------------|----------------------|---------------------------|
> | UltraRE      | OBC     | LR        | 775.22            | 0.4294               | 0.4345                    |
> | RecEraser-LR | BKM     | LR        | 1,319.61          | 0.4230               | 0.4278                    |
> | RecEraser    | BKM     | ATT       | 1,319.61          | 0.4234               | 0.4281                    |
> | SISA-LR      | BRD     | LR        | 1,052.74          | 0.4255               | 0.4273                    |
> | SISA         | BRD     | AVG       | 1,052.74          | 0.4063               | 0.4025                    |
>
> Based on the presented tables, we obtain several key observations:
> * Comparing UltraRE, RecEraser-LR, and SISA-LR, we observe that a lower inertia is associated with a higher recommendation performance. This suggests a positive correlation between clustering performance and recommendation performance.
> * LR demonstrates similar recommendation performance to ATT, surpassing AVG, when compared to the LR-versions (RecEraser-LR and SISA-LR) and their original versions (RecEraser and SISA). This finding is consistent with the results from the "Effect of Combination" section.
> * In ML-1M and ADM datasets, SISA-LR achieves comparable recommendation performance to RecEraser-LR, while outperforming it in ML-100k. This aligns with the differences in inertia across the various datasets, indicating that (i) clustering performance generally correlates positively with recommendation performance, and (ii) the advancements in RecEraser are primarily attributed to the effectiveness of the combiner (ATT).
>
> In conclusion, superior clustering performance corresponds to improved recommendation performance. Our proposed clustering algorithm (OBC) exhibits substantial superiority over the compared methods.
>
> **Q3**:	What contributes to the improvements of UltraRE?
>
> **Response**: For model utility improvement, it is attributed to the superior clustering performance of OBC (as shown in *Response to Q2*). For efficiency improvement, it is attributed to both LR and OBC (as shown in *Response to Q2* in the rebuttal for all reviewers).

---

> > ### Comment · Reviewer_55Fw · 2023-08-21
> >
> > Thanks for the response. Most of my concerns were addressed, I will keep my positive rating.

---

### Official Review · Reviewer_zuBB · 2023-07-04

**Soundness:** 3 good
**Presentation:** 2 fair
**Contribution:** 2 fair
**Rating:** 4
**Confidence:** 4

**Summary:**

This paper addresses the problem of recommendation unlearning, which arises due to privacy concerns and the right to be forgotten. The authors identify limitations in previous methods, which prioritize unlearning efficiency over preserving model utility and fail to optimize the balance between collaboration and balance. To address these limitations, the authors propose a new framework called UltraRE, which simplifies and powers RecEraser for recommendation tasks. UltraRE optimizes the equilibrium between collaboration and balance, ensures convergence on group data, and simplifies the combination estimator. The authors conduct extensive experiments on three real-world datasets and demonstrate that UltraRE outperforms the state-of-the-art recommendation unlearning framework, RecEraser, achieving an improvement in recommendation accuracy and unlearning efficiency on ML-100k, ML-1M, and ADM, respectively.

**Strengths:**

- The paper well-formalize machine unlearning into a three-target problem including unlearning completeness, unlearning efficiency, and model utility.

- The proposed method cleverly transform KMeans into the Monge-Kantorovich problem, to enable optimization for the clustering process.

- The proposed UltraRE performs superior in model utility as well as unlearning efficiency.

**Weaknesses:**

- The paper only focuses on exact unlearning, and ignore the important research line of approximate unlearning [1-2], which is much more efficient when dealing with multiple unlearning requests.
- Compared to the SOTA method RecEraser, the proposed UltraRE only makes incremental improvements, including modifying the KMeans clustering algorithm and applying simpler aggregation method, which makes the technical contribution rather limited.
- The experimental datasets are all of small size, which may limit the accuracy of efficiency test.

[1] GIF: A General Graph Unlearning Strategy via Influence Function

[2] GNNDelete: A General Strategy for Unlearning in Graph Neural Networks

**Questions:**

- How can the proposed method maintain efficient when dealing with multiple deletion request at one time?
- What and how significant is the major technical contribution of UltraRE compared to  RecEraser?

**Limitations:**

The paper targets at solving negative societal impact of existing works.

---

> ### Author Rebuttal · Authors · 2023-08-09
>
> We sincerely thank you for your valuable comments and suggestions. We hope our response addresses your concerns.
>
> **Q1**:	Discussion about approximate unlearning.
>
> **Response**: First of all, it is important to highlight that, in this paper, we focus on exact recommendation unlearning, which is orthogonal to the studies you mentioned, i.e., approximate graph unlearning, in the field of machine unlearning. Exact unlearning approaches fully guarantee the most fundamental requirement of unlearning, i.e., completeness. Algorithmic retraining has been established as the sole authoritative way of ensuring completeness [37]. This is the inherent advantage of exact unlearning and cannot be attained through approximate unlearning. We do agree that approximate unlearning approaches have gained much attention due to their efficiency, and we did not ignore them. Instead, we have reviewed various representative studies in the “Related Work” section. We sincerely apologize for missing the studies you mentioned, and we will ensure to include them during revision.
>
> **Q2**:	How to maintain efficiency when dealing with multiple requests?
>
> **Response**: From a theoretical perspective, ensemble retraining frameworks can increase shard number and employ parallel training. In this way, the time required for unlearning is limited to sub-model retraining. Besides the unique authority and inherent advantages of exact recommendation unlearning that we mentioned in *Response to Q1*, it is noteworthy that within the context of recommendation, the approximate unlearning approaches are not as efficient as expected. This inefficiency can be attributed to two main reasons. First, the inherent computation of the Hessian matrix is time-consuming. Although approximations can be made to accelerate the computation process, these approximations compromise the completeness and model utility, and do not eliminate the need for at least one offline computation of the Hessian matrix. Second, both user and item embeddings are dense feature matrices that contain a large number of parameters, thereby further increasing computational overhead in the context of recommendation.
>
> From a practical perspective, current experiments do not sufficiently demonstrate the efficiency enhancement, because the majority of time is spent during stage II (independent training) while the enhancements occur in stages I (non-overlapping division) and III (model combination). Thus, following [6], we further conduct experiments on ML-10M (large-scale dataset) with 50 shards (large shard number), and report the results in the table below.
>
> | ML-10M    | DMF-RecEraser | DMF-UltraRE | LightGCN-RecEraser | LightGCN-UltraRE |
> |-----------|---------------|-------------|--------------------|------------------|
> | Stage I   | 872.53m       | 259.18s     | 879.06m            | 257.83s          |
> | Stage II  | 213.55s       | 208.44s     | 860.12s            | 852.32s          |
> | Stage III | 83.74s        | 67.26s      | 384.50s            | 376.57s          |
> | Total     | 877.48m       | 534.88s     | 899.80m            | 1,486.72s        |
>
> The results indicate that our proposed UltraRE significantly improves efficiency compared to RecEraser. Specifically, in stages I and III, UltraRE demonstrates average efficiency improvements of **20,227.87%** and 13.30% respectively. Note that, in the large-scale dataset (ML-10M), our proposed clustering algorithm (OBC) significantly outperforms the BKM used in RecEraser, *reducing clustering time from several hours to just a few minutes* (in stage I, 872.53m/879.06m vs 259.18s/257.83s).
>
> **Q3**: The experimental datasets are small-size, which may limit the accuracy of efficiency test.
>
> **Response**: As responded above, following SOTA [6], we have conducted experiments on ML-10M (the largest dataset used in [6]). The experimental results demonstrate higher efficiency improvements when compared to the results reported in the paper, providing additional evidence of the substantial efficiency enhancements achieved by our method.
>
> **Q4**: Major technical contribution compared with RecEraser (SOTA).
>
> **Response**: Please refer to *Response to Q1* in the rebuttal to all reviewers.

---

> ### Author Response · Authors · 2023-08-16
> **A kindly request for your response**
>
> We hope this message finds you well. We are writing to kindly request an update on the status of your response for our rebuttal. As the deadline for discussion draws near, we would greatly appreciate if you could let us know whether the rebuttal addresses your concerns. Thanks for your attention!

---

> ### Comment · Area_Chair_yHNS · 2023-08-21
> **Reminder: Response to the authors' rebuttal**
>
> Dear Reviewer zuBB,
>
> I hope this message finds you well. We would like to remind you about your response to the authors' rebuttal. The authors have submitted their rebuttal, and we are eagerly awaiting your response to proceed with the final decision.
>
> Your expertise and input are highly valued, and we kindly request you to review the authors' rebuttal and share your thoughts as soon as possible. This will enable us to maintain the review timeline and provide timely feedback to the authors.
>
> The deadline for your rebuttal is August 21. Thank you for your time and dedication to the review process.
>
> AC

---

### Official Review · Reviewer_oK8f · 2023-07-08

**Soundness:** 3 good
**Presentation:** 3 good
**Contribution:** 3 good
**Rating:** 7
**Confidence:** 5

**Summary:**

This paper investigates the problem of recommendation unlearning, with a specific focus on the exact unlearning approach. The authors adopt an ensemble-based perspective to redefine the exact unlearning framework and break down the framework into three components regarding prediction error. The primary modification made by the authors pertains to the first component, where a novel optimal balanced clustering algorithm is proposed. The authors also simplify the third component to improve efficiency. Extensive experiments are conducted across three benchmark datasets.

**Strengths:**

1) The investigated problem is interesting. Unlearning has gained increasing attention in the field of privacy-preserving machine learning. This paper focuses on a practical scenario of unlearning, i.e., recommendation system.

2) This paper offers ease of comprehension. The ideas within are clearly presented.

3) This paper uses the theory of ensemble learning to provide a theoretical foundation for the exact unlearning approach. This provides valuable insights for improving the performance of the exact unlearning approach.

4) The authors propose a novel clustering algorithm which achieves an optimal trade-off between (i) shard balance and (ii) sample similarity. This modification effectively tackles  the trade-off issue from an optimization perspective.


**Weaknesses:**

1) The improvement brought by the proposed optimal balanced clustering algorithm appears to be inconsistent. On the one hand, the inertia is significantly reduced. On the other hand, the model utility is not significantly improved. Please explain this phenomenon.

2) There exist some small errors, for example, in line 211, the complexity is O(Nk) instead of O(N^2).


**Questions:**

The improvement brought by the proposed optimal balanced clustering algorithm appears to be inconsistent. On the one hand, the inertia is significantly reduced. On the other hand, the model utility is not significantly improved. Please explain this phenomenon.

**Limitations:**

The authors have discussed the issue that the efficiency improvement of stages I and III is not significant, and provided a practical experimental setting.

---

> ### Author Rebuttal · Authors · 2023-08-09
>
> We sincerely thank you for your valuable comments and suggestions. We hope our response addresses your concerns.
>
> **Q1**:	Inconsistent improvements between inertia and model utility.
>
> **Response**: Inertia is the summation of the inner-cluster distance of all sample-centroid pairs, ranging in $[0, \infty]$. Model utility is evaluated by NDCG and HR, both of which range in $[0, 1]$. Therefore, the scales of inertia and model utility are different, leading to inconsistent improvements between them. To subtly investigate the relationship between inertia and model utility (NDCG@10), we conduct another ablation study. Specifically, we vary the clustering algorithms in stage I (non-overlapping division) and use Logistic Regression (LR) in stage III (model combination) for all compared methods, and report corresponding inertia and NDCG (during learning). We choose LR due to its comparable performance to ATTention networks (ATT), as shown in the "Effect of Combination" section. For easy comparison, we also report the result of original RecEraser and SISA.
>
> | ML-100K      | Stage I | Stage III | Inertia (stage I) | DMF-NDCG (stage III) | LightGCN-NDCG (stage III) |
> |--------------|---------|-----------|-------------------|----------------------|---------------------------|
> | UltraRE      | OBC     | LR        | 816.32            | 0.3847               | 0.3859                    |
> | RecEraser-LR | BKM     | LR        | 1,738.24          | 0.3792               | 0.3808                    |
> | RecEraser    | BKM     | ATT       | 1,738.24          | 0.3795               | 0.3812                    |
> | SISA-LR      | BRD     | LR        | 852.50            | 0.3824               | 0.3836                    |
> | SISA         | BRD     | AVG       | 852.50            | 0.3716               | 0.3684                    |
>
> | ML-1M        | Stage I | Stage III | Inertia (stage I) | DMF-NDCG (stage III) | LightGCN-NDCG (stage III) |
> |--------------|---------|-----------|-------------------|----------------------|---------------------------|
> | UltraRE      | OBC     | LR        | 729.46            | 0.4042               | 0.4241                    |
> | RecEraser-LR | BKM     | LR        | 1,318.31          | 0.3968               | 0.4169                    |
> | RecEraser    | BKM     | ATT       | 1,318.31          | 0.3973               | 0.4171                    |
> | SISA-LR      | BRD     | LR        | 1,129.72          | 0.3972               | 0.4173                    |
> | SISA         | BRD     | AVG       | 1,129.72          | 0.3956               | 0.3941                    |
>
> | ADM          | Stage I | Stage III | Inertia (stage I) | DMF-NDCG (stage III) | LightGCN-NDCG (stage III) |
> |--------------|---------|-----------|-------------------|----------------------|---------------------------|
> | UltraRE      | OBC     | LR        | 775.22            | 0.4294               | 0.4345                    |
> | RecEraser-LR | BKM     | LR        | 1,319.61          | 0.4230               | 0.4278                    |
> | RecEraser    | BKM     | ATT       | 1,319.61          | 0.4234               | 0.4281                    |
> | SISA-LR      | BRD     | LR        | 1,052.74          | 0.4255               | 0.4273                    |
> | SISA         | BRD     | AVG       | 1,052.74          | 0.4063               | 0.4025                    |
>
> Based on the presented tables, we obtain several key observations:
> * Comparing UltraRE, RecEraser-LR, and SISA-LR, we observe that a lower inertia is associated with a higher recommendation performance. This suggests a positive correlation between clustering performance and recommendation performance.
> * LR demonstrates similar recommendation performance to ATT, surpassing AVG, when compared to the LR-versions (RecEraser-LR and SISA-LR) and their original versions (RecEraser and SISA). This finding is consistent with the results from the "Effect of Combination" section.
> * In ML-1M and ADM datasets, SISA-LR achieves comparable recommendation performance to RecEraser-LR, while outperforming it in ML-100k. This aligns with the differences in inertia across the various datasets, indicating that (i) clustering performance generally correlates positively with recommendation performance, and (ii) the advancements in RecEraser are primarily attributed to the effectiveness of the combiner (ATT).
>
> In conclusion, superior clustering performance corresponds to improved recommendation performance. Our proposed clustering algorithm (OBC) exhibits substantial superiority over the compared methods.
>
> **Q2**:	Typos.
>
> **Respond**: We will carefully review the paper and fix any typos during revision.

---

> > ### Comment · Reviewer_oK8f · 2023-08-20
> > **Response to Authors' Rebuttal**
> >
> > Thanks for your response. Your response have addressed my concerns. Please include these additional experimental results in the main content of the manuscript.

---

### Author Rebuttal · Authors · 2023-08-09

We sincerely thank all the reviewers for their valuable comments and suggestions, which are crucial for improving our work. We hope our response addresses your concerns.

**Q1**: Major technical contribution compared with RecEraser (SOTA).

**Response**: We summarize the contributions in the table below, followed by a detailed explanation.

|No.|Contributions|UltraRE (ours)|RecEraser [6]|Significance|
|-|-|-|-|-|
|1|Analyse with error decomposition, aiming to reduce redundancy loss (stage I) and combination loss (stage III)|Theory|Intuition|The first exact unlearning method that is guided by theory|
|2|Propose a novel clustering algorithm (stage I)|OBC|BKM [6, 7, 24]|Improve recommendation performance by 1.28%, reduce time cost by 98.11% (20,227.87% on ML-10M)|
|3|Simplify model combiner (stage III)|LR|ATT|Achieve comparable recommendation performance, reduce time cost by 11.95% (13.30% on ML-10M)|

Firstly and most importantly, we rethink exact unlearning from the perspective of ensemble learning and provide a theoretical analysis for prediction error, which serves as the theoretical basis for existing exact unlearning approaches, i.e., ensemble retraining frameworks. To the best of our knowledge, this is the first exact recommendation unlearning method that offers theoretical guarantees, setting it apart from previous work [6, 24] that relied on the intuition of collaboration preservation. Building upon this theory, we decompose the error into three distinct components and establish correlations with corresponding stages. Our goal is to enhance performance by reducing the error at each stage (I and III). While this may appear similar to the improvement direction pursued by prior works [6, 7, 24], we want to emphasize that our approach is guided by theory, not intuition. In other words, our work provides theoretical support for prior intuitions.

Secondly, we propose a novel algorithm to address an important issue in stage I, i.e., the incongruity between sample similarity and shard balance. Prior work [6, 7, 24] relies on the manual insertion of a priority list to achieve balanced clustering (BKM). This approach, once again, relies solely on intuition without any theoretical support. As shown in our empirical study (Figure 4), BKM fails to effectively resolve this incongruity, compromising sample similarity to achieve shard balance. In contrast, we propose a novel balanced clustering algorithm (OBC) that aims to obtain the optimal solution by optimizing the Monge-Kantorovich Problem. To further enhance efficiency, we solve the optimization objective through Sinkhorn distance. The experimental results (Tables 2 and 3, and Figure 4) demonstrate that the proposed algorithm significantly outperforms compared methods, efficiently achieving an equilibrium between the incongruity. Our freshly included experimental results in *Response to Q2* on a large-scale dataset provide additional evidence of the substantial efficiency enhancements achieved by our method.

Thirdly, guided by the experience in ensemble learning, we simplify the model complexity by employing LR. Additionally, we conduct an empirical study to validate that the model complexity in stage III can be reduced while maintaining comparable performance.

**Q2**: Significance of efficiency improvement.

**Response**: Current experiments do not sufficiently demonstrate the efficiency enhancement, because the majority of time is spent during stage II (independent training) while the enhancements occur in stages I (non-overlapping division) and III (model combination). Thus, following [6], we further conduct experiments on ML-10M (the largest dataset used in [6]) with 50 shards (large shard number), and report the results in the table below.

| ML-10M    | DMF-RecEraser | DMF-UltraRE | LightGCN-RecEraser | LightGCN-UltraRE |
|-----------|---------------|-------------|--------------------|------------------|
| Stage I   | 872.53m       | 259.18s     | 879.06m            | 257.83s          |
| Stage II  | 213.55s       | 208.44s     | 860.12s            | 852.32s          |
| Stage III | 83.74s        | 67.26s      | 384.50s            | 376.57s          |
| Total     | 877.48m       | 534.88s     | 899.80m            | 1,486.72s        |


The results indicate that our proposed UltraRE significantly improves efficiency compared to RecEraser. Specifically, in stages I and III, UltraRE demonstrates average efficiency enhancements of **20,227.87%** and 13.30% respectively. Note that, in the large-scale dataset, our proposed clustering algorithm (OBC) outperforms the BKM used in RecEraser, *reducing clustering time from several hours to just a few minutes* (in stage I, 872.53m/879.06m vs 259.18s/257.83s). The experimental results demonstrate higher efficiency improvements when compared to the results reported in the paper, providing additional evidence of the substantial efficiency enhancements achieved by our method.

---

### Decision · Program_Chairs · 2023-09-21

**Decision:**

Accept (poster)

**Comment:**

The paper introduces an intriguing perspective on the exact unlearning framework, utilizing an ensemble-based approach while incorporating theoretical advancements. The authors' unique integration of theoretical guidance to disentangle the error into three distinct components, each correlating with corresponding stages, is a noteworthy contribution that distinguishes their work from previous intuition-driven modifications to the framework.

The reviews for this submission have generally been positive, with one exception being Reviewer zuBB, who recommended a borderline reject. Upon careful consideration of Reviewer zuBB's critique and the authors' response, it is my opinion that Reviewer zuBB's concern regarding "approximate unlearning" may not carry substantial weight. The authors' response to the other points raised by this reviewer is convincing and demonstrates a robust understanding of the subject matter. Furthermore, it is important to note that Reviewer zuBB has not provided any additional feedback or objections following the authors' response.

Based on my comprehensive evaluation of the paper and the entire review process, I would like to recommend the acceptance for this submission.